# The Effects of Releasing Greenbelt Restrictions on Land Development in the Case of Medium-Sized Cities in Korea

**Jae Ik Kim \*, Jun Yong Hyun and Seom Gyeol Lee**

Department of Urban Planning, Keimyung University, Daegu 42601, Korea; hjy@kmu.ac.kr (J.Y.H.);
ohgongbi@naver.com (S.G.L.)

\* Correspondence: kji@kmu.ac.kr; Tel.: +82-53-580-5278

**Abstract:** Many metropolitan areas around the world aim to control urban growth with a view to achieving efficiency and containing urban problems. Among many urban growth policy tools, the green belt (GB) policy is known as the most rigid and strongest. However, there has been no study on the consequences when GB restrictions are completely removed. The primary purpose of this study is to analyse the spatial effects of greenbelt removal on land development in Korea's medium-sized cities between 2000 and 2017. To do so, we used the Landsat thematic mapper (TM) 5 satellite image (2000) and Landsat OLI TIRS 8 satellite image (2017) along with various attribute data to model the spatial effects of greenbelt removal in the cases of three medium-sized cities in Korea. The result of difference-in-difference (DID) analysis confirms that the effects of GB removal on land development vary depending on the local conditions of land development.

**Keywords:** spatial concentration; greenbelt removal; attribute data; difference-in-difference model; urban growth model; logistic model

---

## 1. Introduction

The spatial concentration of population and economic activities in large cities is perceived as a serious social problem in many developing countries. In most cases, the capacity of urban government to supply proper public services such as public transportation, housing, and other infrastructure is limited compared to the increase in demand accompanied by rapid urbanisation. To overcome urban problems and to achieve efficiency, many metropolitan areas around the world have adopted various containment policies that aim to control urban growth.

Among the kinds of urban containment policies available, a greenbelt (GB), known as the most restrictive form of urban containment policy [1], is a geographical boundary around a city or urban region in which development activity is strongly controlled to prevent urban sprawl. GB policies have been adopted by many cities around world, especially in European and Asian countries [2,3], ever since they were introduced in the Greater London Plan in 1951. However, only a small number of cities have adopted a GB policy in the USA [4]. Instead, many American cities have chosen the urban growth boundary (UGB) as a tool to mitigate urban sprawl. It is designed to attract, rather than prohibit, urban development within the boundary implemented with the urban service boundary, in most cases for a given period.

There has been abundant research on the spatial effects of the urban containment policy. Some studies have tried to prove the effectiveness of the urban containment policy, while others have criticised it. The effectiveness of the policy may stem from the rigidity of the policy. The UGB, as a popular tool of growth management in the USA since the 1980s, encourages new development within

the designated area, while the GB policy strictly prohibits new development within the GB lands. The effectiveness of the policy may also depend on the stage of urban development. At the time of a GB's establishment, the urban boundary generally lies well within the inner side of the GB. Most development occurs within the city. As a city grows, the built-up areas move outward towards the GB. The GB may have policy effectiveness at this stage of urban development, by leading to urban infill development. However, as a city grows further, the effectiveness of the GB becomes uncertain, and depends on the spatial pattern of development.

In Korea, policies to control urban growth were initiated in the early 1960s, when the population began to be concentrated in the Seoul metropolitan area. In the 1970s, several innovative nation-wide growth-control programmes were initiated. Greenbelts, referred to as development-restricted zones, were established around 14 large- and medium-sized cities across the country at the end of July 1971 as a part of the first National Comprehensive Physical Development Plan (1972–1981). The main policy objective of the greenbelt was to prevent unorderly urban expansion, and to protect the natural environment around the city.

The greenbelt policy was one of the strongest and most consistent land conservation policy tools from 1971 to 1999 in Korea. However, there were social tensions between those for and against the policy. Environmental groups strongly supported it while development-oriented firms and land owners criticised it. In particular, those against the greenbelt programme pointed out that the policy causes higher housing and land prices within the urban core by reducing the supply of developable lands, restrictions on private property rights, and other problems.

To lessen the GB-related social problems, the government lifted greenbelt restrictions four times, and significant revisions were made to the policy. For example, the central government formed a Greenbelt Reform Committee in 1999 and released the GB restriction in seven medium-sized cities (1103 km$^2$). At the same time, parts of the GB lands in large cities where environmental value is low were relaxed and made available for development. The GB restriction was valid only in eight large metropolitan cities (Seoul, Busan, Daegu, Daejeon, Gwangju, Ulsan, Sejong, and Changwon) as shown in Figure 1 (more historical background of the Korean GB policy can be found in [1]).

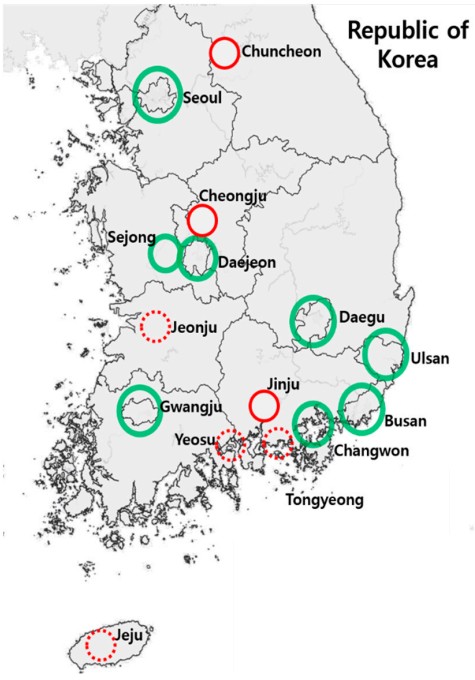

**Figure 1.** Location of greenbelt cities; (1) the greenbelt maintained (green); (2) the greenbelt removed (solid and broken red); and (3) the selected study area (solid red).

More than 15 years have passed since the GB restriction in medium-sized cities was completely removed. That period is long enough to check its effectiveness and to prove the feasibility of the GB policy for medium-sized cities. However, no research attention has been paid to the consequences of the GB removal despite the intense social debate initially. Furthermore, the Korean experience of the GB removal policy may provide useful policy implications for cities where the GB policy has been adopted or its introduction is being considered. For these reasons, this paper examines the spatial effects of the complete removal of GB restrictions on urban land development using the case of Korean medium-sized cities.

## 2. Literature Review

The greenbelt, as a strong policy tool to restrict the physical expansion of built-up areas and to preserve green space for environmental and recreational purposes, decreases the supply of land and thus increases the price of land (and houses). As a result, previous studies of the greenbelt tend to analyse the effects on the housing and land markets [5–11].

There have also been other studies that have focused on the environmental impacts of the greenbelt [12–14], the amenity value of the greenbelt [15–19], and the transportation impacts [20–22].

There has been hot debate on the spatial effects of the urban containment policy. The heart of the debate on the impact of the policy lies in the spatial pattern of development—whether it encourages compact development or brings about leapfrog development. In general, the UGB policy has been considered effective in curbing sprawl by bringing new development within the designated areas [23–26]. The effectiveness of the GB policy, however, is conflicted. Conceptually, the GB leads to densification on its inner side and leapfrog development on its outer side [14,27,28]. Empirically, the effectiveness of the greenbelt as an urban growth management policy is controversial because of its positive and negative effects [1,29]. Some researchers agree that the GB policy results in a spatial concentration of development in core cities, and higher densities [30–32]. Other researchers, in contrast, have pointed out that the GB brought about leapfrog development by shifting urban development beyond the GB area [10,21,22,33,34]. Unlike the UGB, development in GB lands is strictly prohibited. This restriction shifts the location of development to either the inner side or the outer side of the GB. The GB policy is effective when it attracts new development to the city side of the GB. This is extremely difficult when population influx and development pressure are very high. Under this condition, the GB policy alone cannot stop development beyond the GB. For example, Bengston and Youn [1] concluded that Seoul's GB policy has failed to keep development from invading the Capital Region beyond the GB.

Nevertheless, there is an agreement on the spatial effects—the greenbelt affects the shape of cities by surrounding them with a belt of agricultural land or other open space, and imposes a severe restriction on land development, thereby shifting development demand from the greenbelt to other places (the term 'land development' in this article refers to the conversion of land from non-urban to urban use (residential, commercial, industrial, and other uses)).

Recently, some research has focused on the effects of partially relaxing the GB restriction on land development in the case of the Seoul metropolitan area [35,36]. The latter study [36] found that the GB deregulation had significant effects on urban land development near the boundary of the city of Seoul and the GB boundary. Similarly, Han [35] also found that the GB relaxation served to guide new development to the inner areas of the GB while slowing down the rates of development beyond the GB. These studies commonly conclude that land development is more active since the GB release compared with before in the case of partial deregulation of the GB at the edge of the city. This finding must be true because the deregulation was required by high development pressure and was implemented to make land development possible [36,37].

In addition, the GBs of medium-sized cities were removed not because of high development pressure but because of the national policy—'relaxation of unnecessary restrictions'. Therefore, the spatial effects of the GB are expected to be different among cities. In large cities, such as London

and Seoul, urban development is active on both the inner and outer sides of the GB lands. This may lead to compact development if there is plenty of developable lands on the inner side of the GB. Conversely, it may cause leapfrog development if the demand for development is not satisfied in contained areas. However, the development pressure may be relatively low in medium-sized cities, compared to large cities. To date, none of the previous studies have focused on the spatial changes of land development as a result of the removal of the whole GB. This study examines this neglected topic—the effects of the removal of the GB on the spatial distributional shift of urban land development in Korea's medium-sized cities between 2000 and 2017.

This study is distinct in several aspects. First, it analyses the effects of the entire GB abolishment, rather than partial relaxation of the GB lands. Second, it measures the actual impact of the GB removal on land development, rather than providing a counterfactual analysis. Third, it deals with experiences of the GB removal on urban land development in three medium-sized cities, as opposed to the previous large-city case studies.

## 3. Materials and Methods

### 3.1. Data

To measure the effects of GB removal on land development, it was essential to obtain data covering the periods before and after its removal. Since the GB removal started in the early 2000s, the attributes and spatial data before 2000 and after 2017 have been collected.

Attribute data used for this analysis include population data from Statistics Korea (Korea's census bureau) and land price data in 2000 and 2017. Since the data on the market value of land prices were not available, the appraisal land value was used to analyse the policy effect on land value. Land price data were from the Korea Appraisal Board. Distance data (straight-line and network distance data) were calculated by the geographical information systems (GIS) base.

This study utilised Landsat-5 thematic mapper (TM) imagery with 30 m × 30 m spatial resolution for 2000, and Landsat TM 8 for 2015 of the study areas to identify changes in the urbanised land. Administrative district maps were extracted from the national terrain map, and land-use maps (including development-restricted areas, including the GB) were extracted from the Korean land information system.

### 3.2. Study Area

Three medium-sized cities were chosen as study areas considering the locational distribution of the three cites (Table 1 and Figure 1). They represent the central (Cheongju), northern (Chuncheon), and southern regions (Jinju), as shown in Figure 1. These cities had a greenbelt until the early 2000s. Among the three cities, Cheonju has the largest population with the highest population density (921.6/km$^2$), while Chuncheon has the widest city boundary with the lowest population density (302.2/km$^2$).

**Table 1.** Descriptions of the selected GB-released medium-sized cities.

| City Name | Area (2017, km$^2$) | Population (2017) | Population Density | GB Area (km$^2$) | Date of the GB Release |
|---|---|---|---|---|---|
| Jinju | 712.84 | 420,833 | 590.4 | 203.0 | 203.10.31 |
| Chuncheon | 1116.83 | 337,485 | 302.2 | 294.40 | 2001.12.08 |
| Cheongju | 940.33 | 866,648 | 921.6 | 180.10 | 2002.01.19 |

Source: Population Statistics Based on Resident Registration (2017), Statistics Korea [38].

### 3.3. Approach

The effects of the GB removal can be analysed by comparing the amount of land converted from non-urban uses to urban uses in previous GB areas with the amount of developed land in other areas. If the development occurs more actively in the GB areas than in other areas, the GB removal policy is

effective. Likewise, if more people live inside the previous GB lands after the GB removal, the more likely it is that the GB restriction was effective. For this purpose, the study areas were classified into three parts according to their proximity to the city centre (see Figure 2).

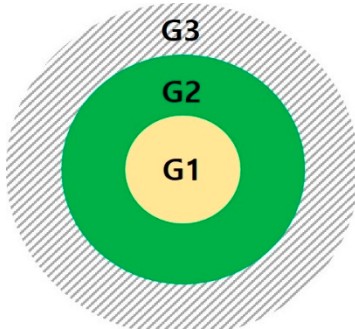

**Figure 2.** Classification of the study area: G1 = inner side of the GB, G2 = GB, and G3 = outer side of the GB.

On the basis of the spatial expansion of actual urban areas, we focus on the changes in previous GB areas. This analysis provides information on the location and quantity of urban development in the absence of a greenbelt policy.

We exclude non-developable lands from the study area. Non-developable lands are defined as lands having environmental hazards or development constraints. These lands include national/local parks, catchment areas, ecological reserves, wild life habitats, dams, and reserves for local plants and species. Lands with physical constraints such as slopes (15 degree or above), altitude (200 m or above), and rivers/streams are also classified as non-developable lands.

To identify the information, multi-spectral satellite images (Landsat TM 5 and TM 8 for 2000 and 2015) of the study area were classified using ERDAS Imagine (software). The image data sets were geometrically corrected to the Universal Transverse Mercator (UTM) coordinate system. Then, these Landsat TM data were merged with the administration boundary maps. All lands were divided into 900 m$^2$ (30 m × 30 m) grid cells for analysis.

*3.4. Model*

In the absence of the greenbelt policy, the land-use change between 2000 and 2017 can be estimated through various modelling approaches. Most studies measure the effect of an urban containment policy with a greenbelt dummy variable in a regression model [34,39]. For example, Bae and Jun [34] estimate the population and employment density gradients with a greenbelt dummy variable in their model to reallocate jobs and workers in order to measure the effects of Seoul's greenbelt on commuting costs.

Many of the previous studies on the effects of relaxing the GB restriction adopt a difference-in-difference (DID) model developed by Abadie [40]. This study also adopts a DID method to examine the effects of the GB removal. The basic DID framework can be expressed as follows:

$$P_i = \beta_0 + \beta_1 D_t + \beta_2 D_{i,t} + \beta_3 (D_t * D_{i,t}) + \gamma X_{i,t} + e_{i,t} \tag{1}$$

where $P_i$ is the probability of development of *i*th individual land at time t (developed = 1, undeveloped = 0), $D_t$ is the time before the greenbelt release ($D_t$ = 1 in 2017, and $D_t$ = 0 in 2000), $D_{i,t}$ is the land parcel within the greenbelt ($D_{i,t}$ = 1, others = 0), and the interaction term $D_t$*$D_{i,t}$ is the actual policy effects of the greenbelt release. $X_{i,t}$ is the vector of socio-economic and land characteristics, and $e_{it}$ is a random error term.

We expect the coefficient of $\beta_3$ to be significant and have a positive sign if the probability of land development in the GB is affected by the GB release.

To estimate policy effects, Han et al. [36] employed the linear probability model of the DID method, while Han [35] utilised the ordinary least squares model of the DID method. We used a logistic regression model rather than a linear probability model because the dependent variable of this study is binary: i.e., whether a grid cell is developed or not during the study period. One of the crucial problems of the logistic regression is spatial autocorrelation, which violates the assumption of independent residuals. The most frequently used method to reduce spatial autocorrelation is reduction of data size through random sampling [41–43]. The sampling makes the analysis easier and computing faster. For the model calibration, five per cent of cells (14,000 cells out of 285,370 cells for Jinju, 19,684 cells out of 393,712 cells for Chuncheon, and 29,586 cells out of 591,782 for Cheongju) were randomly selected from the full data set through the GIS sampling tool. By reducing the number of observations, Moran's I, which measures spatial autocorrelation, was decreased considerably from 0.72, 0.70, and 0.73 at full data to 0.35, 0.28, and 0.32 at sample data. They are all significant with p-values less than 0.001. All models were run in *Stata* version 10.1.developed by StataCorp.

### 3.5. Variables

The basic DID model contains location, time dummy variables, and one interaction term. As there exists the possibility of biased estimates due to self-selection and unobserved heterogeneity, we try to mitigate such problems by employing control variables. Logistic regression models have been used to detect land-use conversion from non-urban land use to urban land use [41,42]. In the logistic regression, the selection of the explanatory variables is data-driven rather than knowledge-driven [42]. Nevertheless, the control variables were selected on the basis of previous studies. The control variables that were used in this study are: (i) distance to city centre, (ii) distance to the nearest main road, (iii) population growth rate, and (iv) rate of land price change. The first two variables reflect the agglomeration factor and are expected to negatively affect land-use conversion [42,43]. The latter two variables represent demand for development. The population growth rate also reflects the density of development.

Proximity to the nearest main road was measured as the straight-line distance because a large portion of the undeveloped parcels is not connected by a transport network. The distance to the city centre was also measured as the straight-line distance. The slope was derived from the 30 m digital elevation maps of the National Geographical Information Institute for the study area. However, we did not include the slope variable in the model partly because its effect on urban development may not be significant as construction technology improves [16,36], and partly because its effect was already reflected when steep slope lands, along with elevation variable, were classified as non-developable lands.

Population growth was selected as the representative socio-economic variable. We assume that the more people live inside the previous GB land after removal of the GB, the more likely the GB restriction was effective. Population growth was calculated on the basis of census data. Another important variable that affects land development is land price. By releasing the GB restriction, more lands are available for development, and thereby development pressures are eased and land price increases are slowed down [35]. To analyse the policy effect on land value, appraisal land value was used because the data on the market value of the land were not available. Parcels within the greenbelt have two layers of land-use restrictions: one is ordinary land use and the other is the greenbelt restriction. We consider the ordinary land-use restriction valid within the greenbelt area after the GB release.

## 4. Results and Discussion

### 4.1. The Effect of the Greenbelt Policy Release on Urban Development

Prior to the analysis of the three individual cities, the regular logistic model that contains city dummy variables was run to identify the city-level effects of development. The maximum likelihood estimator was used to fit the binary logistic regression model. Table 2 shows the estimated coefficients

and odds ratios for the model, which contains five variables and two city dummy variables. The results of the model show several development trends in the city-level effects of the GB. First, the previous GB lands show a higher development probability with an odds ratio higher than 1 (1.644) than do the other lands. Therefore, the removal of the GB played a role in making the lands more attractive to develop. Second, urban development in Cheongju (reference city) was more active than in Jinju, but less active than in Chuncheon. However, the effects of the GB on land development in individual cities reveal quite different results from these city-level analyses.

**Table 2.** The results of logistic model (pooled data).

| Variable | Coefficient | Odds Ratio | *p*-Value |
|---|---|---|---|
| Constant | −2.5649 *** | 0.0770 | 0.000 |
| GB dummy (GB = 1, else = 0) | 0.4969 *** | 1.644 | 0.000 |
| Distance_city centre | 0.0098 *** | 1.010 | 0.004 |
| Distance_main road | −0.003 | 0.9970 | 0.737 |
| Population change (%) | 0.0000 *** | 1.000 | 0.000 |
| Land price change (%) | 0.0050 *** | 1.0050 | 0.000 |
| Chuncheon (dummy) | 0.6034 *** | 1.8284 | 0.000 |
| Jinju (dummy) | −0.2413 *** | 0.7856 | 0.000 |
| LR chi$^2$ | 1385.69 (Prob > chi$^2$ = 0.000) | | |
| Pseudo-R$^2$ | 0.062 | | |
| Log likelihood | −10545.83 | | |

Note: *, **, *** indicate statistically significance at 10%, 5%, and 1% levels, respectively.

The location of land development before and after the release of the GB restriction in the three cities is presented in Table 3 and Figure 3. The table and figure provide some important information on the urban development pattern.

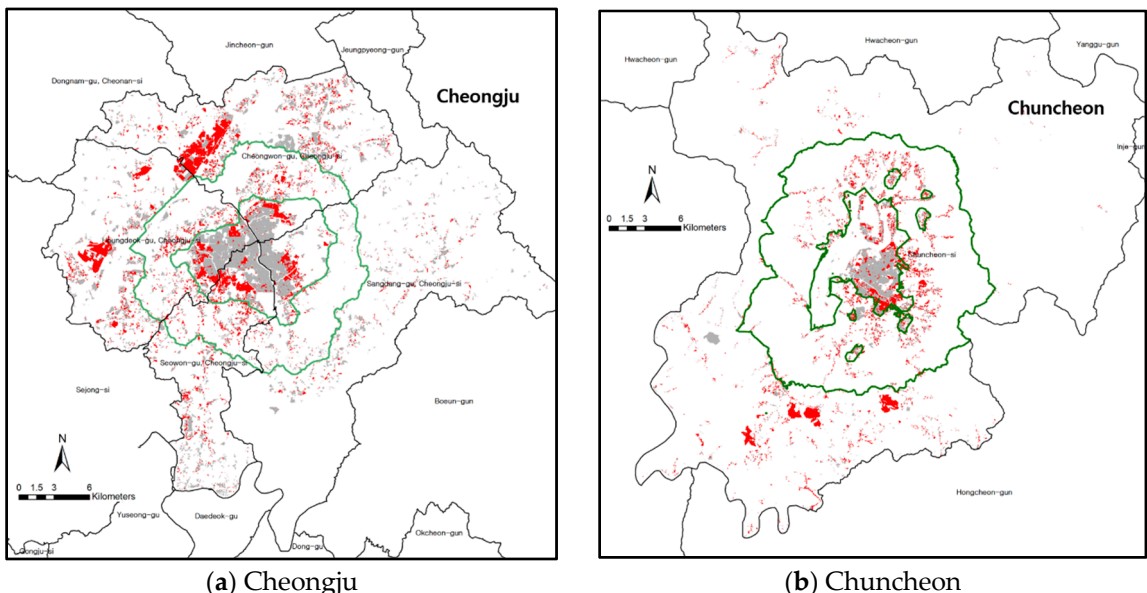

(**a**) Cheongju        (**b**) Chuncheon

**Figure 3.** *Cont.*

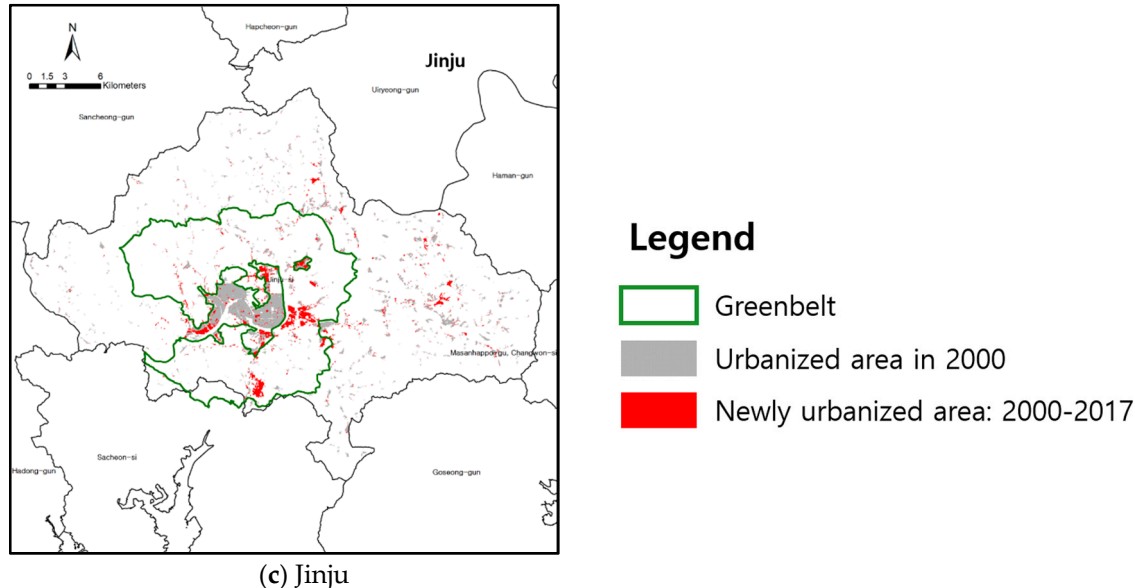

(**c**) Jinju

**Figure 3.** Urbanised areas in Cheongju, Chuncheon, and Jinju: 2000–2017.

**Table 3.** The changes of developed land cells and population.

| | | **(Unit: Persons, %)** | | | | | | | | |
|---|---|---|---|---|---|---|---|---|---|---|
| | | **Jinju** | | | **Chuncheon** | | | **Cheongju** | | |
| | | **2000** | **2017** | **Change** | **2000** | **2017** | **Change** | **2000** | **2017** | **Change** |
| **Population** | **Inner GB** | 279,162 | 287,598 | 3.02 | 236,989 | 265,361 | 11.97 | 578,433 | 657,653 | 13.70 |
| | **In GB** | 87,153 | 92,871 | 6.56 | 54,400 | 57,135 | 5.03 | 57,661 | 49,857 | −13.53 |
| | **Outer GB** | 47,270 | 40,364 | −14.61 | 17,467 | 14,989 | −14.19 | 111,660 | 159,138 | 42.52 |
| | **Total** | 413,585 | 420,833 | 1.75 | 308,856 | 337,485 | 9.27 | 747,754 | 866,648 | 15.9 |
| | | **(Unit: km² , %)** | | | | | | | | |
| **Developed area** | **Inner GB** | 16.55 | 20.90 | 26.29 | 17.55 | 25.51 | 45.38 | 37.06 | 48.19 | 30.03 |
| | **In GB** | 9.71 | 21.62 | 122.61 | 9.36 | 22.85 | 144.08 | 23.16 | 30.24 | 30.58 |
| | **Outer GB** | 23.13 | 31.67 | 36.94 | 8.99 | 27.03 | 200.58 | 57.86 | 88.95 | 53.73 |
| | **Total** | 49.39 | 74.19 | 50.21 | 35.90 | 75.39 | 110.00 | 118.08 | 167.38 | 41.75 |

There were plenty of developable lands on the inner and outer sides of the GB in Cheongju and Chuncheon cities at the time of the GB removal. The developable lands in the contained lands (between urban core and the GB) were 26.2 km² in Cheongju and 21.8 km² in Chuncheon. The existing urbanised areas were 37.1 km² and 17.6 km², respectively, as shown in Table 3. Even if the GB restrictions were removed, the new development occurred on the inner (G1) and outer sides (G3) of the GB, leaving most of the GB areas (G2) undeveloped in the cases of Cheongju and Chuncheon. In contrast, there was little developable land (11.9 km²) on the inner side of the GB in the case of Jinju (16.6 km² of urbanised area) where new development has actively occurred on the previous GB lands.

These different patterns of development before and after GB removal can further be supported by the changes in developed lands and population by area. Overall, Table 3 shows that the growth rate of urbanised areas is faster than that of the population for all three cities. This implies that there has been active non-residential land development such as manufacturing, commercial, and public use in the cities. It shows that each of the three cities reveals distinctive development patterns. In the case of Jinju city, the GB removal leads to urban development through infill. The population has reduced on the outer side of the GB (G3), while that of the inner side and inside the GB (G1) has increased. Urban development is most active in the GB area (G2). The city of Chuncheon reveals a similar pattern of development. However, land development is most active on the outer side of the GB and most of the population increase occurs on the inner side. The urban development of Cheongju city shows the

opposite pattern. Even though the GB has been removed, the population inside the GB has decreased, while the population on the outer side has increased. The amount of land development inside the GB is relatively small compared with that in the inner and outer sides. In other words, GB removal has triggered more active new suburban development. During the study period, one additional person was found to bring about 3421 m$^2$ of new land development in Jinju, 1379 m$^2$ in Chuncheon, and only 414.7 m$^2$ in Cheongju. This finding conflicts with the findings of other studies, which maintain that the GB brings about a pattern of leapfrog development. The amount of developable lands inside the contained area may be a crucial factor that determines the pattern of the urban development as well as the effectiveness of the GB policy [14].

Tables 4 and 5 display the estimated coefficients for the basic DID model (Model-I) containing the three independent variables and Model-II containing the four covariates (two accessibility variables, population change, and land price change). The regression coefficients are as follows.

**Table 4.** Estimation results of the three cities (Model-I).

|  | Jinju | | Chuncheon | | Cheongju | |
|---|---|---|---|---|---|---|
|  | **Coef.** | **Odds Ratio** | **Coef.** | **Odds Ratio** | **Coef.** | **Odds Ratio** |
| **Constant** | 4.4009 *** | 81.52 | −5.7835 *** | 0.0031 | 4.8302 *** | 125.24 |
| **Time** | 2.7706 *** | 15.97 | 4.1238 *** | 61.80 | 2.0409 *** | 7.70 |
| **GB** | −11.6495 *** | 0.000 | −5.1121 *** | 0.060 | −10.2548 *** | 0.000 |
| **DID (time * GB)** | 5.5949 *** | 269.05 | 1.1744 *** | 3.24 | 0.1687 | 1.18 |
| **Observations** | 14,000 | | 19,684 | | 29,586 | |
| **Log likelihood** | −6026.72 | | −10,067.09 | | −13,293.74 | |
| **Pseudo-R$^2$** | 0.015 | | 0.047 | | 0.029 | |

Note: *, **, *** statistically significant at 10%, 5%, and 1% levels, respectively.

**Table 5.** Estimation results of the three cities (Model-II).

|  | Jinju | | Chuncheon | | Cheongju | |
|---|---|---|---|---|---|---|
|  | **Coef.** | **Odds Ratio** | **Coef.** | **Odds Ratio** | **Coef.** | **Odds Ratio** |
| **Time** | 1.552 *** | 4.720 | 2.138 *** | 8.484 | 0.968 *** | 2.632 |
| **GB** | −3.054 *** | 0.047 | −2.790 *** | 0.013 | −4.354 *** | 0.013 |
| **DID (time * GB)** | 1.392 *** | 4.022 | 0.255 | 1.290 | −0.191 | 0.826 |
| **Distance to the CBD (km)** | −0.028 | 0.972 | −0.260 *** | 0.771 | −0.324 *** | 0.723 |
| **Distance to the main road (km)** | −0.088 ** | 0.91 | 0.064 ** | 1.066 | −0.237 *** | 0.789 |
| **Population** | 1.514 *** | 4.543 | 0.002 *** | 1.002 | 0.003 *** | 1.002 |
| **Land_price** | 0.281 *** | 1.325 | 0.219 *** | 1.245 | 1.37 *** | 1.147 |
| **Constant** | −6.552 *** | (0.001) | −2.132 *** | 0.119 | 1.846 *** | 6.334 |
| **Rho (ρ)** | 0.979 | | 0.918 | | 0.921 | |
| **Log likelihood** | −4504.289 | | −8646.779 | | −11388.173 | |
| **Pseudo-R$^2$** | 0.3157 | | 0.2276 | | 0.2484 | |

Note: *, **, *** indicate statistically significance at 10%, 5%, and 1% levels, respectively.

On the basis of the results of Model-I, some useful findings can be derived. The signs of coefficients are the same across the three cities, though the intensity varies. All variables show expected signs with statistical significance at a 1% significance level except for the DID variable of Cheongju. They are positive for time and DID, and negative for GB. Therefore, the GB removal induces more development inside the GB for Jinju and Chuncheon, but not for Cheongju. The odds ratios (OR) of the DID variables of the three cities also support these development trends. The OR of DID of Jinju is 269.05, and that of Chuncheon is 3.24, implying that the probabilities of development in previous GB lands are 269.05 and 3.24 times higher than in other lands, respectively, by removing the GB restrictions. However, the OR of the DID for Cheongju is not statistically significant.

By adding control variables in Model-II, the explanatory powers of the three models have been highly improved. Since rho (ρ) in Model-II is different from zero for all three city models, the panel

estimator is also different from the pooled data. When the control variables were included in Model-II, the signs of the coefficients were not changed, except for the sign of the DID variable in the case of Cheongju. The focus of this study, the DID variable (Time × GB), shows an interesting result. The coefficient of the Jinju DID is positive and significant, that of Chuncheon is positive and not significant, and that of Cheongju is negative and not significant. The negative sign of the DID variable implies that the GB lands were less likely to have been developed than before the removal of the GB restriction. The non-significant DID variable also suggests that the GB removal may not function as a strong incentive for development. The development activities previously outside the GB were more active in Chuncheon and Cheongju than inside the GB while the activities inside the GB were more active than those of other areas in the case of Jinju. These results can be supported further by the ORs of the model. The OR of the DID variable of Jinju is 4.022 while that of Cheongju is less than 1.0 (0.826) and statistically not significant.

Therefore, it is difficult to conclude that there exists a GB policy effect in Cheongju and Chuncheon. The GB policy is valid only in the case of Jinju where developable lands are scarce on the inner side of the GB. The DID coefficient shows that the GB removal accelerates land development in the previously GB boundary. Variations in effects of the GB removal are the results of different demands placed on the GB. There are considerable differences in land development demand for the GB lands among medium-sized cities.

Distance variables have negative signs, implying that higher proximity to the city centre and main road increases development probabilities. The coefficient of the distance to the central business district (CBD) variable of Jinju is not significant, implying that the location of new developments is not clearly distinct by distance. Also, the distance to the main road variable of Chuncheon is positive owing to the mountainous geographical feature. As expected, population and land price variables show positive and significant coefficients.

Although the GB is primarily designed to restrict land development, its removal does not always stimulate land development inside the GB (G2). Land development has increased in all three cities after the GB removal, but the effects of the GB removal are not the same across cities, as the quantity and location patterns of the land development are different among cities. For example, the GB removal apparently contributed to easing development pressures in both G1 and G3, as implied by active development in GB land (G2) in the case of Jinju. Conversely, the GB removal had no significant effects on land development.

## 4.2. Discussion

The greenbelt policy is still an attractive policy instrument, and many cities around the world maintain the policy or are newly implementing it (Ontario in 2005, and Scotland in 2010). Furthermore, some cities are considering introduction of a GB policy.

Is the greenbelt policy helpful to achieving sustainable development? Or is the greenbelt sustainable? Amati [44] raised the question of whether the GB policy is a useful tool for managing urban growth in the twenty-first century. The merits of the GB are to preserve green areas around city and amenity values, and to restrict the expansion of built-up areas thereby leading to infill development. However, in many cases, the greenbelt policy leads to a leapfrog development that stimulates lengthier commutes and higher car use, which increases the price of housing/land.

The role of the GB on green area protection can also be challenged. The removal of the GB itself is perceived by environmental groups as a sign of the government's giving up the protection of green areas around cities. In fact, the greenbelt may not actually be green. It may contain degraded land, little landscape quality, and limited public access [45]. The GB has an important inter-generational function as a land reservoir for future use as well. Therefore, even though some parts of the GB are not worth protecting for their environmental aspects, the GB's inter-generational function is valid and should be considered. This may call social attention to the time span of GB policy if development pressure is intense in GB lands. For example, there is a fixed time span (usually 20 years) in UGB policy

while the GB is perpetual, once established. How long should the GB be maintained at the expense of the current generation?

Effective land-use regulation contributes to achieving a more efficient form of human settlement, which also improves regional economies [46]. As in the case of introducing new land-use regulations, the removal of the GB also affects the efficiency of the urban structure. It provides more lands for urban uses near existing urbanised areas. If demand for development is high and developable lands are scarce, then it is very likely to lead to compact development. Conversely, if demand for development is weak, the removal of the GB may have no impact.

The effectiveness of the GB policy may depend on many factors, such as the country-specific political climate concerning more development or preservation [47], and the amount of available lands for development [14,48]. The core question in the GB policy may lie in how to maintain a balance between the demands for development and preservation of the lands [47].

## 5. Conclusions

This study compares the current spatial distribution of three medium-sized cities where the whole GB was removed based on the cities' actual experiences rather than counterfactual or quasi-natural experiments. The results of this study show that the effects of the GB removal are not the same across the cities. The policy had a significant effect on urban land development in Jinju, a moderate effect in Chuncheon, and no effect in Cheongju. The effects depend on the characteristics of the city. Conceptually, the GB attracts infill development by prohibiting new development within the GB lands. However, if there is not enough land for development in city-side areas, leapfrog development is inevitable. Variations in the effects of the GB removal of the three medium-sized cities may be the result of different demands placed on the GB. They may depend on the intensity of the development pressure on previous GB lands. If abundant vacant lands exist in the inner city, the GB restriction may not be effective. For example, in the case of Cheongju, land development was more active in non-GB land than GB land after the GB release. In contrast, GB release in a city where developable lands become scarce and thus development pressure intensifies, as in the case of Jinju, very likely leads to infill development by providing more developable lands to the city.

By incorporating the controversial views on the effects of the GB on the location of urban development—infill versus leapfrog development and suggestions from previous studies [14]—we reach the following conclusion. If there are plenty of developable lands between the core city and the GB, then the pattern of urban development tends to be infill development. Conversely, scarce lands between the core and the GB under conditions of high demand for development will lead to leapfrog development. Therefore, we cannot simply conclude that the GB is an effective policy tool to control the location and density of urban development without considering the local conditions of the land market.

This study provides some useful implications for cities that are currently under pressure to develop the GB. The different impacts of GB removal on spatial urban structure may stem from a variety of reasons. If we choose the amount of developable lands in a contained area as a critical criterion of the GB policy's effectiveness, the different effects of its removal in the three cities can easily be interpreted. If there is enough developable land to absorb development pressure on the inner side of the GB, new development on GB lands may not be necessary. In contrast, if the development pressure is intense because of the influx of people and employment, the contained lands are subject to be developed. It is beyond the scope of this study to identify the exact conditions that are necessary to remove or introduce the GB policy. They may be related to economic growth, the speed and ratio of urbanisation, income, or the population growth of a city. More case studies are required to be able to generalise. The changes in the land development pattern may not be due to abandonment of the GB, but because of other socio-economic changes, if the stable unit treatment value assumption (SUTVA) is not satisfied. However, this study confirms that the local conditions of urban development do affect

the effectiveness of the GB policy. Therefore, the establishment, abolishment or even partial removal of the GB must be implemented only after careful planning consideration of local conditions.

**Author Contributions:** This study is a result of collaboration. J.I.K. conceptualised the study idea, developed the research methodology, and wrote most of the paper. J.Y.H. analysed the data with statistical models. S.G.L. prepared attributes and spatial data and visualised the study results.

**Funding:** This research was supported by Basic Science Research Program through the National Research Foundation of Korea (NRF) funded by the Ministry of Education (20170385).

**Conflicts of Interest:** The authors declare no conflict of interest.

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
