# Peer review of "The Effects of Releasing Greenbelt Restrictions on Land Development in the Case of Medium-Sized Cities in Korea"

_sustainability, doi:10.3390/su11030630_

Round 1

Reviewer 1 Report

In general, this is a well written paper on an important subject. However, some major changes are still needed before publication. Please see comments below:

Line 15: I am not sure of this line in the abstract because I have read studies before regarding the famous “removal” of Seoul's greenbelt. For example:

https://www.jstor.org/stable/26267777?seq=1#metadata_info_tab_contents

https://www.fs.usda.gov/treesearch/pubs/13220

I think this paper still has value but the authors need to use another way of framing their originality and contributions in the abstract. I think it should be more directed towards theoretical contributions rather than doing something no one has done before.

Line 39: After this paragraph, I expect to see some discussions on the policy effects and theories of greenbelt policies worldwide. Discussion of Korea’s greenbelt should appear later. It decreases the interests of readers if you head into cases without enough engagement of the general theories and methods.

Line 92: In the literature section, it is important to define which “effects” your research will focus on, and then discuss some debates around it. After reading the current literature review, I am not sure what part of the debate this paper wants to take part in.

Line 178: For DID study I would expect either a IV test for endogeneity. For example, it could be totally possible that land development is driving the socio-economic variables (such as population growth) rather than the GB policy. Another test this paper does not do is assigning other time cut-off to Dt=1 for testing whether 2017 is a significant time cut-off point. I suggest the authors to conduct at least one of the experiments and discuss another (or acknowledge limitations for what are not tested).

Line 200: The selection for socio-economic variables should have some previous literature to back-up. I actually think the literature review for this paper misses the part of discussing policy impacts on land development, for example:

https://www.sciencedirect.com/science/article/pii/S0166046217303472

Line 283: Higher loglike value does not mean better model unless you run a LR test between the 2 models and the results suggest significant improvement.   

Line 326: I suggest the authors to have dedicated result discussion section rather than meshing them in the conclusions. The lack of policy implications and theoretical discussions is one of the reasons that this paper is very short and it is the main weakness of this paper. There are abound previous discussion on how land planning policy affects sustainable or economic development in cities. If you imagine your study in a much larger discussion of how spatial planning policy can be effective, rather than just narrowly focused on an ad-hoc GB discussion, the paper would be much for interesting. Some previous discussion include:

https://journals.sagepub.com/doi/abs/10.1177/0885412210382985?casa_token=nDXvRMoE4JkAAAAA:H0D_bab6mx61aQOSKG-wZ9W69wM9ukWSUblqt2pbQA6W5Wi-a4WOektDYtHPM144n24CE4yvzmiOqw

https://onlinelibrary.wiley.com/doi/full/10.1002/ldr.3106

Author Response

First of all, we want to thank you for your valuable comments and suggestions. We revised our paper as follows.

Point 1. Line 15: I am not sure of this line in the abstract because I have read studies before regarding the famous “removal” of Seoul's greenbelt. For example Bengston and Youn, 2005 and 2006. I think it should be more directed towards theoretical contributions rather than doing something no one has done before.

Response: The main focus of the Bengston & Youn (2005; 2006) was to analyze the costs and benefits of the GB policy in the case of Seoul metropolitan area. Seoul’s GB has never been removed, though small parts of the GB lands (112.5km2 out of 1566.8km2) were relaxed for various reasons. It still exists as explained in manuscript. We agree that theoretical contributions may be more valuable than analyzing real world phenomena. We will seriously consider it as our next task.

Point 2. Line 39: After this paragraph, I expect to see some discussions on the policy effects and theories of greenbelt policies worldwide. Discussion of Korea’s greenbelt should appear later. It decreases the interests of readers if you head into cases without enough engagement of the general theories and methods.

Response: We added some general effects of the urban containment policy after the paragraph in lines 41-51. In addition, more detailed discussions were added in literature review.

Point 3. Line 92: In the literature section, it is important to define which “effects” your research will focus on, and then discuss some debates around it. After reading the current literature review, I am not sure what part of the debate this paper wants to take part in.

Response: We revised the literature review. The primary focus of this paper is to identify the effects of the GB removal on spatial distribution of development. In other words, we want to identify whether the GB lands were developed more actively after the removal policy compared to other lands. This will give us some useful information on the effectiveness of GB policy.  

Point 4. Line 178: For DID study I would expect either a IV test for endogeneity. For example, it could be totally possible that land development is driving the socio-economic variables (such as population growth) rather than the GB policy.

Response: We agree with you. The primary purpose of this paper is to test whether the GB removal attracts more active land development in the previously GB lands than other areas in the case of the three medium-sized cities. We believe that drivers of land development, beside the GB removal, are many, such as socio-economic changes of the cities. Following other researches which estimate urban growth prediction model, we consider the stable unit treatment value assumption (SUTVA) is a reasonable assumption and thus estimates have credibility. We mentioned about this problem in limitation of this study in conclusion (lines 432-434).

Point 5. Line 178: Another test this paper does not do is assigning other time cut-off to Dt=1 for testing whether 2017 is a significant time cut-off point. I suggest the authors to conduct at least one of the experiments and discuss another (or acknowledge limitations for what are not tested).

Response: There is no clear time cut-off to determine the right time interval to check the effects of the GB removal. It is very difficult to derive proper time interval because there is no previous study which analyzes impacts of the complete GB removal. We assume that 15 years after the GB removal is not too short and not too long either, as explained in lines 78-79.

Point 6. Line 200: The selection for socio-economic variables should have some previous literature to back-up. I actually think the literature review for this paper misses the part of discussing policy impacts on land development.

Response: We mentioned some previous literature (i.e., Cheng and Masser, 2003; Poelmans and Van Rompeay, 2010) to back-up for the selection for socio-economic variables in lines 236-245.

Point 7. Line 283: Higher loglike value does not mean better model unless you run a LR test between the 2 models and the results suggest significant improvement.

Response: We deleted the statement because higher pseudo-R2 already proves that the model fit has improved.

Point 8. Line 326: I suggest the authors to have dedicated result discussion section rather than meshing them in the conclusions. The lack of policy implications and theoretical discussions is one of the reasons that this paper is very short and it is the main weakness of this paper. There are abound previous discussion on how land planning policy affects sustainable or economic development in cities. If you imagine your study in a much larger discussion of how spatial planning policy can be effective, rather than just narrowly focused on an ad-hoc GB discussion, the paper would be much for interesting.

Response: We separated the discussion section from conclusion adding more previous studies on the urban containment policy. We wish we will be able to derive effective way of spatial planning policy in near future study.

Reviewer 2 Report

Generally, the research is at a high level and can be published after minor corrections, mainly connected with disucussion section. Introduction  and review of the literature broadly discuss the background of  research, present the research goal and research hypothesis. Research methods have been carefully selected and described. The results of the studies are also clearly presented, although they require small correctiosn indicated below. In my opinion, the part presenting the research results and discussion should be separated. In  the discussion section, it would also be worth comparing the obtained  results with the results of research on greenbelt restrictions in other big cities around the world. You should also give some information of possible errors or limitations of the study and indicate further direction of the study.

Author Response

Point 1. In my opinion, the part presenting the research results and discussion should be separated. In the discussion section, it would also be worth comparing the obtained results with the results of research on greenbelt restrictions in other big cities around the world. You should also give some information of possible errors or limitations of the study and indicate further direction of the study.

Response: We want to thank you for your valuable comments and suggestions. We separated discussion section in MS, adding more experiences of GB policy in other large cities, such as Seoul and London in lines 106-112. Some possible errors were also mentioned in conclusion.

Reviewer 3 Report

The paper is well-conceived and analyses the greenbelt relaxation policy of South Korea.

The literature review is concise, yet the authors should include a more comprehensive literature review to highlight why studying the greenbelt relaxation policy is important. There are two sides to the story on greenbelt: one arguing that the costs of maintaining the greenbelt have exceeded their benefits while the others contending that the environmental benefits of the greenbelt are still substantial. The authors touch on this subject later on in the discussion section, but I think it should be included in the literature review to elucidate the significance of this research.

I feel like having a conceptual model to highlight the overall research design is necessary to make the model easier to understand.

The modeling analysis part is interesting. However, without having a true control group of counterfactual, the model is violating the Stable Unit Treatment Value Assumption (SUTVA) – therefore, failing to establish a true causal inference. I think the authors should include an explanation on how they addressed this issue in their modeling analysis.

The discussion part should include a thorough discussion on the political nature of greenbelt and how the greenbelt relaxation has been at the heart of the political agendas of the past presidents as well as the municipal governments. Discussing the political economy of greenbelt relaxation policy will make the paper interesting.

Finally, the paper has some writing issues: spelling, punctuation, and syntax are incorrect in several places.

Following is the list of references the authors should include in their revisions while addressing the comments.

Amati, Marco, 2008. Green belts: a twentieth-century planning experiment. In: Amati,

Marco (Ed.), Urban Green Belts in the Twenty-First Century. Ashgate, Hampshire,

England, pp. 1–18.

Gordon, David, Scott, Richard, 2012. In: Amati, Marco (Ed.), Ottawa’s Greenbelt Evolves

from Urban Separator to Key Ecological Planning Component. Urban Green Belts in

the Twenty-First Century. Ashgate., Hampshire, England, pp. 129–147.

Hack, Gary., 2012. “Shaping Urban Form.” Planning Ideas That Matter: Livability,

Territoriality. Governance, and Reflective Practice, pp. 33.

Han, Albert T., and Min Hee Go. 2019. “Explaining the National Variation of Land Use: A Cross-National Analysis of Greenbelt Policy in Five Countries.” Land Use Policy 81 (February): 644–56. https://doi.org/10.1016/j.landusepol.2018.11.035.

Kim, Jekook, Kim, Tae-Kyung, 2012. Issues with green belt reform in the Seoul metropolitan

Area. In: Amati, Marco (Ed.), Urban Green Belts in the Twenty-First

Century. Ashgate, Hampshire, England, pp. 37–57

Author Response

Point 1. The paper is well-conceived and analyses the greenbelt relaxation policy of South Korea.  

The literature review is concise, yet the authors should include a more comprehensive literature review to highlight why studying the greenbelt relaxation policy is important. There are two sides to the story on greenbelt: one arguing that the costs of maintaining the greenbelt have exceeded their benefits while the others contending that the environmental benefits of the greenbelt are still substantial. The authors touch on this subject later on in the discussion section, but I think it should be included in the literature review to elucidate the significance of this research.

Response: First of all, we want to thank you for your valuable comments and suggestions. We revised literature review adding more previous researches on the effects of the GB policy. The costs-benefits of the GB is briefly mentioned by introducing the results of Bengston and Youn’s (2006) study (in lines 111-112).

Point 2. I feel like having a conceptual model to highlight the overall research design is necessary to make the model easier to understand.

Response: When revising this paper, we are aware of making the research design simple and easy to understand, especially in introduction and model section.

Point 3. The modeling analysis part is interesting. However, without having a true control group of counterfactual, the model is violating the Stable Unit Treatment Value Assumption (SUTVA) – therefore, failing to establish a true causal inference. I think the authors should include an explanation on how they addressed this issue in their modeling analysis.

Response: We agree with your comments. We mentioned about the possible SUTVA problem when explaining limitation of this study (lines 432-434).

Point 4. The discussion part should include a thorough discussion on the political nature of greenbelt and how the greenbelt relaxation has been at the heart of the political agendas of the past presidents as well as the municipal governments. Discussing the political economy of greenbelt relaxation policy will make the paper interesting.

Response: We separated discussion section from conclusion. Since our primary purpose is to identify the spatial effects of urban development, rather than the political or socio-economic driving forces of the GB removal, we briefly mentioned about such factors in lines 394-396.

Point 5. Finally, the paper has some writing issues: spelling, punctuation, and syntax are incorrect in several places.

Response: This paper is edited by professional English editor.

Point 6. Following is the list of references the authors should include in their revisions while addressing the comments.

Response: Thank you for your list of reference. We added some of them in MS, especially in literature review and discussion section.

Reviewer 4 Report

This study aims to test the effects of releasing greenbelt restrictions on land development by using three medium-sized cities in Korea as the case. The change of greenbelt restriction policy in Korea does provide a promising opportunity to study the effectiveness of urban growth management policy. However, I do not think the current manuscript provides rigorous analysis and useful findings for the rest of the world to learn from due to the following concerns.

The first concern is lack of controlling variables. The current proximity, population growth, and land price are good controlling variables, but there are so many other factors that could affect whether a piece of land would be developed or not.  For instance, the amount of available lands or vacancy rate of buildings in the inner cities could affect new developments. The authors need to conduct a literature on this and add more controlling variables. Additionally, I think the change of land price (instead of land price, the same as population growth instead of population) would be a better control variable.

The second concern is the way how the three cities are handled. Currently, a separate model is estimated for each individual city, which losses the opportunity of testing city level effects. I think a model including all the three cities is needed. There are two potential ways to do that. First, add a dummy variable for each city and run a regular logistic model. Second, run a multilevel logistic model. Due to the small sample of cities (three), the first way might be more appropriate. Either way, the variables at the city level are needed to test the city level effect.

The third concern is the conclusions are overreached, partially due to the first two concerns. “The results of this study show that the effects of the GB removal are not the same across the cities” (line 329), this is the only conclusion supported by the current analysis. All other conclusions are without support. For example, the authors say “the effects depend on the characteristics of the city” (Line 331), there is no any characteristics at city level that has been tested. Another example, the authors say “if there are plenty of developable lands between the core city and the GB, then the pattern of urban development tends to be infill development” (Line 334), there is no variable measuring the availability of developable lands that has been tested.

Here are some other minor comments.

This statement has been made several times in the manuscript – a greenbelt (GB) is the most restrictive form of urban containment policy. Evidence and/or references are needed to support this statement.

The definition of greenbelt policy should be stated at the beginning of the introduction and made clear.

After sampling, are these Moran’s I statistics still significant? Please report the z-score.

Why only three out of seven cities with GB restriction removal were selected? How were they selected?

What year of the area is for in Table 1?

Table 2: the change of population density is another interesting variable to take a look. That might provide some evidence for whether the removal of GB restriction cause the change of density.

The model results are actually consistent with what the data shows in Table 2. In the models, DID is only significant for Jinju. In Table 2, the changes of developed area in Outer GB for Chuncheon and Cheongju is the highest.

Author Response

This study aims to test the effects of releasing greenbelt restrictions on land development by using three medium-sized cities in Korea as the case.

Point 1. The first concern is lack of controlling variables. The current proximity, population growth, and land price are good controlling variables, but there are so many other factors that could affect whether a piece of land would be developed or not. For instance, the amount of available lands or vacancy rate of buildings in the inner cities could affect new developments. The authors need to conduct a literature on this and add more controlling variables. Additionally, I think the change of land price (instead of land price, the same as population growth instead of population) would be a better control variable.

Response: We are grateful to your valuable comments. We appreciate your kind suggestions on our paper again. And we have made all the corrections according to your suggestion. We agree that the amount of available lands or vacancy rate of buildings in the inner cities could affect new developments. We wish we could use them as explanatory variables. Unfortunately, the two variables are very difficult to quantify because we estimate the logistic models with panel data. Instead, we mentioned about the importance of the two variables and explained more back-ups for variable selection in lines 232-240.

We are sorry to create confusion on the variables. They are the rates of change rather than absolute amount of changes. We added more explanation to clarify them in line 247 as well as in Table 2.

Point 2. The second concern is the way how the three cities are handled. Currently, a separate model is estimated for each individual city, which losses the opportunity of testing city level effects. I think a model including all the three cities is needed. There are two potential ways to do that. First, add a dummy variable for each city and run a regular logistic model. Second, run a multilevel logistic model. Due to the small sample of cities (three), the first way might be more appropriate. Either way, the variables at the city level are needed to test the city level effect.

Response: We pooled the three city data and estimated coefficients with regular logistic model with city dummy in lines 261-273. The model gives city level effects. The results were shown in Table 2.

Point 3. The third concern is the conclusions are overreached, partially due to the first two concerns. “The results of this study show that the effects of the GB removal are not the same across the cities” (line 329), this is the only conclusion supported by the current analysis. All other conclusions are without support. For example, the authors say “the effects depend on the characteristics of the city” (Line 331), there is no any characteristics at city level that has been tested. Another example, the authors say “if there are plenty of developable lands between the core city and the GB, then the pattern of urban development tends to be infill development” (Line 334), there is no variable measuring the availability of developable lands that has been tested.

Response: We revised conclusion, based on both supported by current analysis and by linking them with findings of previous researches.

Point 4. Here are some other minor comments.

This statement has been made several times in the manuscript – a greenbelt (GB) is the most restrictive form of urban containment policy. Evidence and/or references are needed to support this statement.

Response: The GB policy is known as the most restrictive and rigid urban containment policy. We added one study, among many previous studies, as an evidence of the statement in lines 32-33.

The definition of greenbelt policy should be stated at the beginning of the introduction and made clear.

Response: We stated a definition of the GB defined in other studies in lines 33-34.

After sampling, are these Moran’s I statistics still significant? Please report the z-score.

Response: Moran’s I statistics for the three cities are 0.35, 0.28, and 0.32. The z-scores are 326.9, 337.8, and 548.5, respectively. They are all significant with p-value less than 0.001. These are reported in lines 226-227.

Why only three out of seven cities with GB restriction removal were selected? How were they selected?

Response: We considered locational distribution of the three cites – Northern (Chuncheon), Mid (Cheongju), and Southern (Jinju) part of the South Korea, as shown in the Figure 1. The logic of city selection was described inn lines 157-159.

What year of the area is for in Table 1?

Response: The administration boundary has not been changed during study period. We show the year (2017) of area in the Table 1.  

Table 2: the change of population density is another interesting variable to take a look. That might provide some evidence for whether the removal of GB restriction cause the change of density. The model results are actually consistent with what the data shows in Table 2. In the models, DID is only significant for Jinju. In Table 2, the changes of developed area in Outer GB for Chuncheon and Cheongju is the highest.

Response: We agree with you. We implicitly reflect the effect of density with the change of the population in the model, keeping in mind that population change is not applicable for non-residential grid cells.

Round 2

Reviewer 1 Report

The paper is fine but the written and language needs improvement. I will leave it to the editor for this part

Reviewer 3 Report

The author has responded adequately to my previous comments. I will advise the authors to work a little bit more on improving the quality of the writing. 

Reviewer 4 Report

I am satisfied with the authors’ responses to my previous comments. I think the paper has been improved significantly.